

# Ear Defender
## Web Crawler for detecting audio DeepFakes

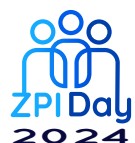

**Autors**: Łukasz Jałocha⬤ · Łukasz Janiak⬤ · Tymoteusz Zapała⬤

**Supervisor:** Piotr Syga⬤

### Abstract

The proliferation of Audio DeepFakes has enabled the creation of highly convincing yet deceptive audio, leading to critical threats such as identity theft and the spread of misinformation. As DeepFake generation tools become increasingly accessible, coupled with the ease of sharing manipulated files across social media and other platforms, the global impact of these threats is escalating. To address this challenge, we present EarDefender, a web-based application designed to detect and combat audio DeepFakes. EarDefender uses cutting-edge AI models, SSL-wav2vec, and MesoNet, to ensure robust detection of manipulated audio. The platform scans the chosen Internet segment, verifies audio authenticity, and generates detailed summary reports, enabling users to identify and mitigate DeepFake risks easily, regardless of technical expertise. Through rigorous evaluation, the application has demonstrated high performance in delivering fast and reliable detection. EarDefender marks a significant advancement in safeguarding online spaces from the dangers of audio manipulation.

## 1 INTRODUCTION

Rapid advances in artificial intelligence have led to the development of fake audio, a sophisticated form of voice manipulation that can mimic human speech with alarming precision. DeepFake technology has become increasingly accessible, with modern tools such as ElevenLabs[1] and Voice.ai[2] allowing users to generate convincing audio manipulations with minimal input, requiring as little as an authentic audio sample and a personal computer. This democratization of technology has raised concerns about its potential for misuse.

Audio DeepFakes, much like their video counterparts, pose a significant and growing threat globally. From July 2023 to July 2024, over 38 countries reported incidents of public figures being impersonated through DeepFake technology [4]. While video DeepFakes tend to dominate public attention, audio DeepFakes are equally critical, often serving as complementary tools to enhance the believability of manipulated visuals. In particular, some spoofing techniques rely solely on audio, emphasizing its standalone impact. Compounding this issue is the fact that misinformation spreads 70% faster than accurate information on platforms such as Twitter [1], increasing the reach and influence of DeepFakes. This highlights the urgent need for robust solutions to counteract their spread, particularly as such content circulates rapidly through social media and other online spaces.

### Project Objectives

In response to these challenges, we developed EarDefender, a web-based application designed to combat the proliferation of DeepFake audio. The primary objective of EarDefender is to empower individuals and organizations to verify the authenticity of audio content found online. The application provides the ability to analyze audio from specific websites and delve into linked or referenced content, offering comprehensive validation across web segments. Our project objectives are as follows.

**High Reliability**: EarDefender uses cutting-edge technologies to achieve robust and accurate Deep-Fake detection. Given the current technological limitations in generalizing across different languages and audio types, which will be discussed in the section 2, our goal is to integrate models capable of achieving a mean equal error rate (EER) of 0.3 or lower across various test datasets. These benchmarks are feasible for state-of-the-art models and sufficient for reliable DeepFake detection, particularly since the application performs detailed analyses of audio, segment by segment.

**Fast Analysis**: The two key time-consuming elements in each analysis are verification by the model and browsing and downloading by the scraper. Ensuring efficient performance is critical. The system must ensure that the analysis of an individual audio file by the model does not exceed the duration of

---

[1] `https://elevenlabs.io/`, accessed: 2024-11-25.
[2] `https://voice.ai/`, accessed: 2024-11-25.

the file itself. The second objective pertains to the time required to find and download the audio, which is more challenging to estimate. This is due to its strong dependence on factors such as the quality of the internet connection, the number and length of files discovered online, and the number of pages visited (as each page requires a few seconds to load its content). While we cannot specify an exact time frame within which the scraper must complete its work, we aim to ensure that its processing time remains proportional to the number of visited pages, the number of files retrieved, and the length of those files. By meeting these two objectives, we can guarantee that the total analysis time for a session, encompassing all files being processed, remains proportional to the combined duration of those files, without exceeding the aggregate time.

**Multiple Analysis Support**: The application must support concurrent analysis sessions, ensuring that each session adheres to the analysis time constraint per analysis. For multiple sessions, the total processing time needs to scale proportionally.

**Compatibility with Popular Platforms**: To maximize its impact, EarDefender is being developed to operate across the most widely used social media platforms that allow audio uploads and support content accessibility by unauthenticated users. Success will be measured by the application's ability to reliably analyze audio files from at least three major platforms, such as YouTube, Facebook, and TikTok.

We believe that these goals position EarDefender as a reliable and efficient solution for combating DeepFake audio, offering robust detection capabilities, fast performance, and compatibility with major platforms. By achieving these objectives, EarDefender aims to empower users with the tools needed to verify audio authenticity across diverse online environments. The codebase of our application is available here[3].

## 2   RELATED WORK

The rapid advancement of DeepFake audio technologies in recent years has underscored the urgent need for effective detection solutions. Due to the field's relative novelty, progress in DeepFake detection has predominantly stemmed from two main sources: academic research and software applications developed by industry leaders.

Significant progress in DeepFake detection has been driven by academic research, particularly through initiatives like the recurring ASVspoof Challenges [7]. Organized biennially since 2015, these challenges are spearheaded by leading universities and research institutions, and have become a cornerstone for advancing the field of audio spoofing detection. The ASVspoof Challenges are initiatives that provide benchmarks for assessing and improving state-of-the-art techniques, showcasing cutting-edge methodologies and architectures. Each iteration brings advances, propelling the field closer to robust and generalized solutions. Despite these efforts, achieving consistent generalization across diverse datasets and audio types remains a significant hurdle [5, 8].

A primary focus of DeepFake detection research is the development of deep neural networks with various architectures. These networks aim to return a range of evaluation results that balance accuracy, efficiency, and generalizability. For example, SSL-wav2vec, introduced during the 2021 ASVspoof Challenge [10], has set a benchmark as a state-of-the-art model. Its ability to generalize effectively across multiple datasets has made it a key technology in our project [9]. This foundational model serves as a cornerstone of EarDefender, leveraging its strengths for reliable and scalable DeepFake detection.

Currently, the market offers a limited range of software solutions dedicated to detecting DeepFake audio. Notable examples include Reassemble.ai[4] that provides a suite of audio-related functionalities, including cloning, synthesis, translation, and DeepFake detection, and Deepfake Detector[5] and AI Voice Detector[6] that focus exclusively on audio verification.

Although these solutions address the core challenge of DeepFake detection, they share several common limitations.

- Only one audio file can be analyzed at a time.

- Results are limited to binary decision "DeepFake" or "not DeepFake" determination for each file.

---

[3]`https://github.com/tymem12/ear-defender`
[4]`https://www.resemble.ai/`, accessed: 2024-11-25.
[5]`https://deepfakedetector.ai/`, accessed: 2024-11-25.
[6]`https://aivoicedetector.com/`, accessed: 2024-11-25.

· Users must manually upload audio files from their local systems for analysis.

These constraints present significant usability challenges, particularly for non-technical users. For example, the process of downloading audio files from online platforms and reuploading them for analysis is cumbersome and often impractical. Such limitations highlight a critical gap in user experience, which EarDefender aims to address.

## How EarDefender Stands Out

EarDefender introduces an innovative approach that directly addresses the shortcomings of existing solutions. Instead of relying on locally stored audio files, EarDefender operates directly on web-based content. Users can provide a URL where the audio resides, eliminating the need for downloading and reuploading files. This streamlined process enhances accessibility and usability, particularly for non-technical audiences.

What truly sets EarDefender apart is its expanded analytical scope. Unlike other tools that focus solely on the specified audio file, EarDefender evaluates all audio files on the given web page and extends its analysis to linked or referenced pages. This capability enables the comprehensive validation of multiple audio files in a single session, saving time and effort.

In addition to broadening the analytical scope, EarDefender provides detailed insights that surpass basic binary classification. Instead of merely indicating whether an audio file is a DeepFake, it pinpoints specific segments within the file that exhibit DeepFake characteristics. This granular feedback is invaluable in scenarios where only portions of the audio are manipulated or when background noise or low-quality recordings might lead to misclassifications. Users can see whether a file is classified as a DeepFake and identify the exact segments that contribute to this classification. This level of precision fosters user confidence and improves the overall utility of the tool.

By addressing common challenges in usability, analytical depth, and accessibility, EarDefender represents a significant advancement in the field of audio DeepFake detection.

## Design Assumptions and Constraints

The EarDefender project was developed with a focus on key architectural and technological considerations to ensure both functionality and modularity. a fundamental design decision was the adoption of a microservices architecture, which allowed for modular development and a clear division of responsibilities. This approach enabled the creation of four distinct modules: the Detector module, tasked with predicting the authenticity of audio files; the Scraper module, responsible for browsing the internet and downloading audio files; the Frontend module, which provides an intuitive user interface; and the Connector module, ensuring seamless communication between other modules and with the database. Communication across these modules is facilitated through the HTTP protocol, enabling standardized and efficient data exchange.

A notable feature of the detector module is the integration of two complementary prediction models. The first, SSL-wav2vec, leverages state-of-the-art self-supervised learning techniques for high accuracy in audio analysis tasks. The second model, MesoNet, uses Linear Frequency Cepstral Coefficients (LFCC) as input and, while less accurate than SSL-wav2vec [9] [5], offers significantly faster performance due to its smaller parameter size. This makes MesoNet an effective lightweight alternative, particularly for tasks that require rapid analysis. These models were rigorously evaluated on diverse datasets, and the results are detailed in section 3.2. The Scraper module was developed using libraries such as yt-dlp to enable audio downloads from specific websites. In addition, Selenium was used to support real-time web scraping, increasing the ability of the module to handle dynamic and varied website structures.

Despite its achievements, the development of EarDefender encountered several limitations. The team consisted of only three members instead of the standard four, which limited the available resources for development. Furthermore, the project was completed under tight time restrictions, which curtailed opportunities for extensive fine-tuning of the models and the addition of more functionalities.

# 3   RESULTS

We present you the results achieved during the development of EarDefender, highlighting the functionalities implemented, the technical objectives met, and the metrics that demonstrate its success. The focus is on the platform's capabilities, its ability to meet user needs effectively, and its potential for practical application in combating audio DeepFakes.

## 3.1 Core functionalities

**Initiating Audio Analysis**

The primary functionality of EarDefender revolves around initiating audio analyses. Users begin by providing a starting URL and, optionally, adjusting specific parameters. The application systematically visits websites, downloads audio files, and analyzes them using AI models. Thanks to its implementation, users receive real-time results as files are processed. This approach eliminates the need to wait for an entire analysis to complete in order to see the results for part of the files, significantly improving efficiency. Users have the flexibility to customize several parameters to tailor the analysis process:

- Chosen Model: Select the specific AI model to be used for analysis.

- Search Depth: Define how deeply the application should explore subpages from the starting URL.

- Maximum Pages: Set the upper limit of pages the application should visit.

- Maximum Number of Files: Specify the maximum number of audio files to download and analyze.

- Maximum Time of Analysis: Impose a time constraint for the overall analysis process to prevent excessive resource usage.

- Maximum Time Per File: Limit the time spent downloading any single file to avoid processing overly lengthy audio files.

These customizable options allow users to optimize resource usage and focus on relevant data, making the platform highly adaptable to diverse requirements

**Analysis History**

EarDefender offers a robust and secure user account system, enabling users to create and manage accounts using their login credentials. This system ensures that users can securely store and access their previous analyses, maintaining a comprehensive history of all completed tasks. By revisiting past results without the need for reprocessing the same data, users save valuable time and computational resources.

To facilitate detailed exploration of past work, EarDefender provides users with the ability to browse their analysis history in depth. For each analysis, the platform displays links to all analyzed files for quick reference, timestamps indicating when the analyses were conducted, the number of files flagged as DeepFakes, and the total number of files found during the analysis.

Additionally, EarDefender delivers detailed file-level insights for every analyzed file. This includes graphical representations that highlight specific segments of audio identified as DeepFakes. Leveraging AI models that process 4-second audio fragments, the application evaluates each file segment by segment, offering a granular view of flagged portions. This detailed approach enables users to pinpoint precisely which parts of the audio exhibit DeepFake characteristics, ensuring a thorough understanding of the analysis results.

**File Prediction Optimization**

To enhance efficiency, EarDefender stores model predictions for every analyzed file. If the same file is encountered during a different user's analysis, the application retrieves the stored results instead of reprocessing the file. This optimization ensures that popular audio files are analyzed only once, reducing redundant computations and improving processing speed for users.

**Extensible and Robust Detection Module**

At the heart of EarDefender lies the Detector Module, which serves as the application's most critical component. Its design prioritizes adaptability and performance, ensuring users have access to cutting-edge detection capabilities now and in the future.

To support this vision, EarDefender employs abstract implementation patterns, making it highly extensible. This architecture facilitates the seamless integration of new models as advancements in detection technologies emerge. By ensuring the application remains adaptable, EarDefender provides a future-proof solution that evolves alongside the latest innovations in audio analysis and detection.

Complementing this extensibility is the inclusion of a dedicated REST API for Model Evaluation. It allows for systematic evaluation against predefined datasets (as detailed in section 3.2), ensuring that key metrics are transparent and reproducible, enabling everyone to validate the results and achieve consistent outcomes when using the same datasets.

## 3.2 Technical objectives results

To ensure that the technical objectives of EarDefender were met, we conducted a series of tests focusing on its core functionalities. These tests served as the primary criterion for determining whether the project achieved its intended goals.

### High Reliability

The Detector Module forms the backbone of EarDefender, as its accuracy directly influences the platform's ability to identify DeepFake audio reliably. To evaluate the models used within the module, we followed established methodologies, primarily leveraging well-known datasets designed for ASVspoof Challenges. The main metric used for the evaluation was the Equal Error Rate (EER) [6], i.e., a value where False Positive Rate and False Negative Rate are equal, a standard measure in DeepFake detection tasks.

We selected three publicly available datasets commonly used in DeepFake detection research:

- In_the_wild [8]- It consists of 37.9 hours of audio clips that are either fake (17.2 hours) or real (20.7 hours) and feature English-speaking celebrities and politicians. Currently, it is the most well-known dataset on which the evaluation is performed. Most of the files are 3- 8 seconds long.

- Deep_voice [3]- This dataset contains examples of real human speech alongside DeepFake versions generated using retrieval-based voice conversion techniques. It was selected for its focus on longer audio clips (up to 2 minutes), which closely resemble the types of DeepFakes commonly encountered online, such as political monologues or interviews.

- MLAAD_v3 [9]- This dataset consists of DeepFake audio generated in eight different languages using a diverse set of text-to-speech (TTS) models. For our evaluation, we used a subset featuring Polish DeepFakes, which includes 2 hours of audio with clip lengths typically between 3 and 12 seconds. This dataset allowed us to test the performance of the model in languages less commonly evaluated.

In addition to these public datasets, we created two custom datasets to reflect real-world conditions more accurately:

- test_eng- This dataset includes 0.7 hours of popular DeepFake audio in English, accompanied by an equal amount of authentic audio of similar length. The DeepFakes were selected from widely circulated audio clips, ensuring that they represented mainstream examples.

- test_pol- This dataset includes 1.0 hours of DeepFake audio and 0.8 hours of real audio. The samples were chosen to reflect realistic Internet audio, often longer than 60 seconds and containing features such as multiple speakers, periods of silence, and sometimes background noise. This dataset aimed to evaluate the models in challenging real-world scenarios.

To test the Detector Module, we evaluated three different models:

- SSL-wav2vec: a state-of-the-art model, which was not fine-tuned for this evaluation.

- MesoNet with LFCC (Trained on ASVspoof 2019): This version of MesoNet uses Linear Frequency Cepstral Coefficients (LFCC) as input and was trained on the ASVspoof 2019 dataset without additional fine-tuning.

- MesoNet with LFCC (Fine-Tuned on ASVspoof 2021): This variant of MesoNet was fine-tuned on the larger ASVspoof 2021 dataset, which includes more comprehensive and diverse examples of DeepFake audio. Fine-tuning was performed to assess whether additional training on larger DeepFake datasets could enhance its predictive capabilities.

The evaluation process assesses whether each audio segment is correctly classified as real or fake, rather than determining whether an entire file is classified as a DeepFake. The latter approach would require setting a custom threshold to label a file as a DeepFake, introducing potential bias. By treating each segment as an independent test instance, we significantly increase the size of the testing examples, leading to more robust and reliable evaluation metrics.

The testing process focused on two key metrics: the Equal Error Rate (EER) as primary metric and accuracy. Accuracy was also calculated, especially for datasets composed solely of DeepFake audio (where EER cannot be computed), its interpretation requires caution due to potential biases introduced by unbalanced class distributions [6].

As shown in Table 1, the following observations were made:

Table 1: Comparison of EER and ACC results from Mesonet, SSL-wav2vec, and finetuned Mesonet.

| Dataset | Mesonet + LFCC | | SSL-wav2vec | | Mesonet + LFCC (finetuned) | |
|---|---|---|---|---|---|---|
| | EER | ACC | EER | ACC | EER | ACC |
| Test_eng | 0.45247 | 0.61968 | 0.34500 | 0.66442 | 0.46805 | 0.68232 |
| Test_pol | 0.41136 | 0.40308 | 0.37481 | 0.60674 | 0.41998 | 0.31179 |
| deep_voice | 0.44166 | 0.81818 | 0.30000 | 0.69090 | 0.50000 | 0.80909 |
| in_the_wild | 0.57027 | 0.36475 | 0.13333 | 0.76814 | 0.54160 | 0.42050 |
| MLAAD_v3_pl | - | 0.98798 | - | 0.72274 | - | 0.99742 |

**In_the_wild dataset**: The EER and accuracy values for both MesoNet models were below the random guessing threshold, highlighting poor predictive capabilities for this dataset. These results are not surprising, as lightweight models often struggle to achieve high performance on this challenging dataset [5, 8]. The SSL Wav2Vec model demonstrated significantly better results, with an EER of 0.13333 and accuracy of 0.76814. These metrics suggest that the model was able to effectively differentiate between real and fake audio on this balanced dataset, with performance comparable to the results reported by other researchers [9].

**English datasets (Test_eng and Deep_voice datasets)**: These datasets, which share similarities in structure, returned comparable results for the SSL Wav2Vec model. The EER for these datasets was slightly above 0.3000, which, while not optimal, is satisfactory for completely custom datasets. The MesoNet models demonstrated lower EER and higher accuracy for the Deep_voice dataset, likely due to a bias towards the majority class (DeepFakes), explaining the high accuracy values despite poorer generalization.

**Polish datasets (Test_pol and MLAAD_v3_pl)**: For the Polish datasets, the accuracy obtained by SSL Wav2Vec on the Test_pol dataset was lower than that achieved on MLAAD_v3_pl [9]. This difference may indicate that SSL Wav2Vec struggles more with languages for which it has not been explicitly trained or fine-tuned.

To summarize, the SSL Wav2Vec model was the only one to meet the defined goals, achieving a mean EER below 0.3 and a mean accuracy slightly below 0.7 across the evaluated datasets. These results demonstrate its potential as a reliable detection tool, particularly for high-quality audio files. In contrast, the MesoNet models, both original and fine-tuned versions, failed to meet the specified constraints and did not perform well overall. Fine-tuning the MesoNet model on the ASVspoof 2021 dataset did not produce significant improvements. This is likely due to the similarity between the ASVspoof 2019 and ASVspoof 2021 datasets, which may have caused the model to overfit on ASVspoof-style data, leading to poor generalization on non-ASVspoof challenge datasets. These findings suggest that while SSL-Wav2Vec shows promise for further development, lightweight models like MesoNet require substantial refinement or alternative approaches to achieve competitive performance in diverse real-world scenarios.

## Time analysis

To ensure that our application meets the required time constraints for processing audio files, we conducted two tests: one focused on the functionality of the detector module and the other on the performance of the scraper module.

First test to measure the analysis time for different models. The evaluation involved analyzing 1,000 files from the MLAAD_v3_pl dataset, with audio lengths ranging from approximately 1 to 18 seconds. Each file was analyzed separately using the SSL Wav2Vec and MesoNet models.

Table 2: Model Analysis Times and Audio Length Statistics. Tests were performed in the dockerized environment.

| Metric | Mean (s) | Std (s) |
|---|---|---|
| **Mesonet** | 0.173 | 0.063 |
| **Wav2Vec** | 3.639 | 1.210 |
| **Audio Lengths** | 7.345 | 3.088 |
| **30s Analysis (Mesonet)** | 0.707 | 0.258 |
| **30s Analysis (Wav2Vec)** | 14.863 | 4.960 |

As shown in Table 2, the SSL Wav2Vec model is significantly slower than the MesoNet model. There is

a strong correlation between the analysis time for the SSL Wav2Vec model and the length of the audio file. Longer files are divided into multiple 4-second segments, with each segment undergoing individual evaluation, leading to increased processing time. Despite this, the time constraint—requiring that the analysis time does not exceed the duration of the file itself—was met. This demonstrates that while the SSL Wav2Vec model is slower, it remains feasible for practical use cases within the defined performance limits.

The second test focused on measuring the performance of the scraper module in terms of time required for web crawling and downloading audio files. As mentioned in Section 3.2, several factors can influence the time needed for these tasks. To analyze these factors, we conducted a series of tests using different parameters, that influence number of downloaded files, the number of visited pages, and the total duration (in seconds) of the downloaded audio files. Each test was repeated five times to calculate the mean and standard deviation of the scraping times.

| Number of Files | Number of Pages | Audio Duration [s] | Mean Scraping Time [s] | Std [s] |
|---|---|---|---|---|
| 1 | 1 | 231 | 14.3 | 0.9 |
| 2 | 2 | 390 | 26.8 | 1.5 |
| 3 | 3 | 749 | 41.2 | 2.4 |
| 3 | 31 | 369 | 237 | 10.9 |

Table 3: Summary of scraping results

As shown in Table 3, the results indicate that the number of visited pages has a significant impact on the total scraping time, more so than the number of downloaded files. This is likely because the scraper pauses for 5–10 seconds on each page to allow the content to load fully, which adds considerable time when the number of pages increases. The duration of the audio files appears to have a lesser influence on the scraping time. This is likely due to the relatively short duration of the files (2 to 4 minutes), which does not contribute significantly to the overall time compared to the waiting time per page. Additionally, the quality and speed of the internet connection can also impact the scraping time. Despite these influencing factors, the results are satisfactory. Even for analyses involving a large number of pages and files, the scraping times remain reasonable and do not pose a significant bottleneck for broader analyses.

### Compatibility with Popular Platforms

To assess the effectiveness of the scraper, we conducted a final test using audio files from three major platforms: YouTube, TikTok, and Facebook. For each platform, the scraper successfully performed the intended analysis, confirming its ability to reliably handle and analyze audio files from widely used social media platforms, as required to meet the objectives of EarDefender

## 4 SUMMARY AND CONCLUSIONS

The development and evaluation of EarDefender demonstrated that most of the project's primary goals were successfully achieved. The application integrated the SSL Wav2Vec model, which met the essential requirement of achieving a sensible Equal Error Rate (EER) while maintaining acceptable analysis times. Specifically, the model achieved a mean EER below 0.3 and a mean accuracy slightly below 0.7 across the evaluated datasets, highlighting its potential as a reliable detection tool, particularly for high-quality audio files. Additionally, the SSL Wav2Vec model was able to analyze a 30-second audio file in approximately 15 seconds, a performance well-suited for a model of its size, satisfying the defined requirements for analysis time.

The application was also tested across three major platforms - YouTube, Facebook, and TikTok - confirming its compatibility with popular social media services. Scraping processes were found to be efficient and proportional to the number of web pages visited and the length of downloaded audio files. The overall time for scraping was reasonable, demonstrating that the module can handle web-based audio sources effectively.

EarDefender represents a significant step forward as the first application of its kind capable of predicting audio authenticity directly from web sources rather than relying solely on locally stored files. While the project has achieved impressive results in terms of performance and compatibility, there are still areas for improvement to further enhance its scalability and prediction capabilities.

## Future Directions

The continuous evolution of DeepFake detection technologies necessitates ongoing development and refinement of EarDefender. Several key areas have been identified for future improvement:

**Integration of State-of-the-Art Models**: The primary focus should remain on maintaining the platform's cutting-edge performance by integrating the latest state-of-the-art models. Continuous research and evaluation will be essential to ensure the tool adapts to the evolving landscape of DeepFake generation techniques.

**Language-Specific Fine-Tuned Models**: To enhance prediction accuracy, future development could involve creating model variations fine-tuned for specific languages. This approach would involve implementing a Spoken Language Identification (SLI) model [2] to determine the language of the input audio. Based on the output, the appropriate language-specific model would be applied for analysis. This strategy could significantly improve detection accuracy, especially for non-English audio.

**Improving Multi-User Support**: Addressing the current limitation of user concurrency should be a priority. Optimizing the application to handle multiple simultaneous users, potentially by leveraging cloud-based infrastructure, would greatly enhance its scalability. This improvement would make the tool more practical for larger user bases, expanding its potential reach and usability.

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
