# OpenReview forum: "Web Crawler for detecting audio DeepFakes"
_pwr.edu.pl/Wrocław_University_of_Science_and_Technology/2024/ZPI_Day — Wrocław University of Science and Technology 2024 ZPI Day Submission_

### Official Review · Reviewer_2tyu · 2024-12-04
**Wysokiej jakości artykuł oraz projekt**

**Confidence:** 4
**Significance Of Results:** 5
**Overall Quality:** 5

**Compliance With Template:**

5: Very High Quality – The article contains all the required sections, which are written in a very detailed, clear, and error-free manner. The structure is professional and meets expectations, and the content adheres to the highest substantive and formal standards.

**Description Of Results:**

4: High Quality – The results are described in detail and supported by usage examples or evaluations. The description is reliable but may lack full depth of analysis.

**Feedback On Consistency:**

Artykuł napisany bardzo dobrze. Słownictwo oraz budowa zdań na bardzo wysokim poziomie. Spójność i czytelność pracy jest bardzo wysoka.
Brak obrazków przedstawiających opracowany system.

**Potential For Development:**

Projekt jest w stanie nadającym się do bezpośredniej komercjalizacji. Dalszy rozwój wskazany, ze szczególnym naciskiem na autorskie rozwinięcia algorytmów wykrywania DeepFake'ów.

**Project Nature Evaluation:**

Praca spełnia w całości wymogi stawiane pracom inżynierskim. Przedstawione rozwiązanie jest nowatorskie oraz wysokiej jakości. Zastosowane technologie dobrze dobrane do projektu. Autorzy wykazują wysoki poziom wiedzy oraz wyczucia inżynierskiego.

**Technical Language Precision:**

5: Very High Quality – The language is entirely appropriate for a technical report. All terms are used correctly and precisely, and the style is professional, clear, and coherent, without any errors or ambiguities.

---

### Official Review · Reviewer_kZsA · 2024-12-05
**EarDefender, ZPI 2024 review**

**Confidence:** 5
**Significance Of Results:** 5
**Overall Quality:** 5

**Compliance With Template:**

5: Very High Quality – The article contains all the required sections, which are written in a very detailed, clear, and error-free manner. The structure is professional and meets expectations, and the content adheres to the highest substantive and formal standards.

**Description Of Results:**

5: Very High Quality – The results are described in detail, clearly and comprehensively, supported by thorough evaluation, analysis, and convincing usage examples. The description meets the highest substantive standards.

**Feedback On Consistency:**

The paper is consistent with the project description. All required sections are included and follow the logical order. The language used is precise, and most claims are supported by appropriate references.
The tables are referred to in the text and properly captioned. The authors show an experimental, numerical evaluation of their work (even though the results are not good enough to be publishable in the current version, it is important to note that those results are reported on a dataset scrapped directly from the internet). Additionally, for practical reasons and hinting at potential deployment in a real-world environment, the authors show the time efficacy of their work.
On a technical note, table 3 is captioned below instead of above. Tables 2 and 3 indicate seconds as a unit of measure in different brackets.

**Potential For Development:**

The authors indicate the main directions for the future, describing each with several sentences, showing that the idea is thought through.

**Project Nature Evaluation:**

The project is an engineering work as described in sections 2 and 3, as the authors propose a tool for internet scrapping and DeepFake detection done with previously developed models that were integrated into the app.
The article states claims that are verifiable, showing obtained results. Moreover, it is important to note that the authors show their codebase, making the claims possible to verify.

**Technical Language Precision:**

5: Very High Quality – The language is entirely appropriate for a technical report. All terms are used correctly and precisely, and the style is professional, clear, and coherent, without any errors or ambiguities.

---

### Official Review · Reviewer_pccm · 2024-12-07
**Ear Defender-Web Crawler for detecting audio DeepFakes**

**Confidence:** 2
**Significance Of Results:** 4
**Overall Quality:** 4

**Compliance With Template:**

4: High Quality – The article contains all the required sections, which are well-written and substantively correct, although minor errors or shortcomings may be present. The overall structure is clear and coherent.

**Description Of Results:**

3: Average Quality – The results are described with moderate detail. Some examples or evaluation elements are present but insufficiently developed or incomplete.

**Feedback On Consistency:**

- The problem analysis is clear and well-structured, but the link between the identified problem and the chosen methodology could be strengthened to ensure coherence.
- The results are presented effectively, but additional explanation of the figures would help clarify their significance.
- The conclusions are consistent with the results but could more explicitly connect back to the original problem statement.

**Potential For Development:**

- the project that clearly identifies open questions or propose specific extensions for their work demonstrate potential for development.
- the project that contributes foundational tools or theories exhibit significant developmental potential.

**Project Nature Evaluation:**

- The project identifies a clear industrial need and offers practical solutions, emphasizing its high utility.
- The focus on theoretical exploration limits the immediate practical application of the results.

**Technical Language Precision:**

4: High Quality – The language is appropriate for a technical report. Terminology is used correctly, and statements are precise, with only minor shortcomings that do not affect the overall clarity.

---

### Decision · Program_Chairs · 2024-12-10

Accept (Oral)